# Emerging Fungal Infections: New Patients, New Patterns, and New Pathogens

**DOI:** 10.3390/jof5030067

**Published:** 2019-07-20

**Authors:** Daniel Z.P. Friedman, Ilan S. Schwartz

**Affiliations:** Division of Infectious Diseases, Department of Medicine, Faculty of Medicine & Dentistry, University of Alberta, Edmonton, AB T6G 2G3, Canada

**Keywords:** aspergillosis, candidiasis, *Candida auris*, antifungal resistance, endemic mycoses, epidemiology, *Emergomyces*, *Blastomyces*, *Sporothrix brasiliensis*, invasive fungal disease

## Abstract

The landscape of clinical mycology is constantly changing. New therapies for malignant and autoimmune diseases have led to new risk factors for unusual mycoses. Invasive candidiasis is increasingly caused by non-albicans *Candida* spp., including *C. auris*, a multidrug-resistant yeast with the potential for nosocomial transmission that has rapidly spread globally. The use of mould-active antifungal prophylaxis in patients with cancer or transplantation has decreased the incidence of invasive fungal disease, but shifted the balance of mould disease in these patients to those from non-fumigatus *Aspergillus* species, Mucorales, and *Scedosporium/Lomentospora* spp. The agricultural application of triazole pesticides has driven an emergence of azole-resistant *A. fumigatus* in environmental and clinical isolates. The widespread use of topical antifungals with corticosteroids in India has resulted in *Trichophyton mentagrophytes* causing recalcitrant dermatophytosis. New dimorphic fungal pathogens have emerged, including *Emergomyces*, which cause disseminated mycoses globally, primarily in HIV infected patients, and *Blastomyces*
*helicus* and *B. percursus*, causes of atypical blastomycosis in western parts of North America and in Africa, respectively. In North America, regions of geographic risk for coccidioidomycosis, histoplasmosis, and blastomycosis have expanded, possibly related to climate change. In Brazil, zoonotic sporotrichosis caused by *Sporothrix brasiliensis* has emerged as an important disease of felines and people.

## 1. Introduction

In clinical mycology, as in other facets of healthcare, the only thing constant is change. Medical advances have improved and prolonged lives, but have also increased the pool of individuals vulnerable to fungal disease; among these, new therapies for old diseases—such as monoclonal antibodies for autoimmune disease and small-molecule inhibitors (e.g. receptor tyrosine kinase inhibitors like ibrutinib) for B-cell malignancies—have resulted in reports of atypical and unusually severe fungal infections [1]. At the same time, more sophisticated methods to identify fungi have led to the recognition of genetic and phenotypic diversity among fungal pathogens [2]. Finally, decades of prolonged antifungal and antibacterial use in agriculture and medicine have altered the global microbiome, with a consequence being the emergence of drug-resistant fungal infections of plants, animals, and humans [3]. Within these contexts, changing trends are explored in global epidemiology of human fungal infections.

## 2. New Patients at Risk for Fungal Disease

### Immunotherapy: Medical Progress Begets New Risk Factors for Invasive Fungal Disease

Immunotherapies have revolutionized the treatment of cancers and autoimmune diseases. Their infectious risks, however, are only beginning to be fully appreciated. Invasive fungal infections are important complications of some of these novel immunomodulators. Notably, the Bruton’s tyrosine kinase inhibitor, ibrutinib, used to treat B-cell malignancies, is associated with severe and unusual fungal infections, particularly with *Aspergillus* and *Cryptococcus* [4,5]. Fingolimod, a syphingosine-1-phosphate receptor used for the treatment of relapsing-remitting multiple sclerosis, has been identified as a possible risk factor in some patients who developed cryptococcosis and histoplasmosis [6,7,8]. 

Cell cycle checkpoint inhibitors, such as inhibitors of CTLA4, PD1 and PD-L1, are used for several different cancer types (particularly melanoma, non-small cell lung cancer and hematologic malignancies), and, due to the upregulation of the immune system, are hypothesized to have antifungal properties [9]. However, both invasive aspergillosis and candidiasis have been reported following the use of the checkpoint inhibitor, nivolumab, for the treatment of non-small cell lung cancer [10]. As their use continues to increase, more studies are required to better define the infectious risk, particularly for opportunistic fungal infections, with these novel immunomodulators.

## 3. Emerging Yeast Infections

### 3.1. Shift to Non-albicans Candida Species and the Emergence of Antifungal Resistance

Invasive candidiasis is a serious infection that primarily affects critically ill and immunocompromised patients. *Candida albicans*, usually susceptible to fluconazole, has long been the most prevalent species implicated in both invasive and mucocutaneous infections. However, with increasing use of antifungal agents and new diagnostic techniques, a change in the epidemiology of *Candida* infections is occurring, with the consequence of increased incidences of infections caused by species with less predictable antifungal susceptibility. 

Established in 1997, the SENTRY Antifungal Surveillance Program monitors the global epidemiology of invasive *Candida* infections with respect to species distribution and resistance to antifungals. Most recently, the program has published its data from the first 20 years, which included over 20,000 clinical isolates collected through passive surveillance from 39 countries worldwide. Although *C. albicans* remains the most prevalent species causing invasive candidiasis, the total proportion of infections attributable to *C. albicans* has decreased from 57.4 to 46.4% over the 20-year surveillance period [11]. In the United States, over 30% of cases of candidemia are now caused by *C. glabrata,* a concerning trend given the increased rates of antifungal resistance associated with this species [12]. In some centers, resistance to echinocandins—mediated by mutations to the *FKS1* hot spot—is seen in as many as 10% of *C. glabrata* isolates, and among these echinocandin-resistant strains, azole co-resistance occurs in 20% [13]. Conversely, in the United Kingdom, *C. glabrata* isolates were rarely resistant to echinocandins (0.55%) [14]. Also more frequently encountered is *C. parapsilosis*. This species is attributable for 15% of cases of candidemia in the United States [12], 20% in Russia [15], and rivals *C. albicans* as the leading cause of invasive disease in South Africa [16]. The rise of *C. parapsilosis* may be a marker of poor infection control practices, because *C. parapsilosis* is a commensal of the skin and can be transmitted within healthcare settings. This is worrisome because *C. parapsilosis* is also commonly resistant to antifungals. As many as 60% of *C. parapsilosis* isolates were fluconazole-resistant in South Africa [16].

*Candida* species have undergone a taxonomic reclassification, with the emergence of new clades, genera and species over the last decade. Stavrou et al. systematically reviewed the literature of epidemiology of yeasts species, identified by molecular techniques causing invasive *Candida* infections [17]. Although in 13 epidemiologic studies from 2013–2018 *C. albicans* remained the most prevalent agent of candidiasis, newly-reclassified yeasts were recognized, including species of the genera *Kluveromyces*, *Pichia*, and *Wickerhamiella*. This cladisitic analysis resulted in species groupings of similar antifungal susceptibility patterns, emphasizing the importance of proper identification to facilitate appropriate empiric therapy.

### 3.2. The Global Emergence of Candida auris

One of the most troubling changes in the epidemiology of invasive candidiasis is the worldwide emergence of *C. auris*, a multidrug resistant organism potential for efficient nosocomial transmission. Since it was described in Japan in 2009 [18], *C. auris* has been reported from 32 countries from six continents (Figure 1) [19]. In an examination of 54 isolates by whole genome sequencing, Lockhart et al., showed that there are four phylogeographic clades of *C. auris* from three continents. The average genetic distance between isolates within the clades was <70 single-nucleotide positions (SNPs), whereas between the clades isolates differed by 20,000 to 120,000 SNPs [20]. This suggests that *C. auris* emerged simultaneously in different corners of the globe, with a subsequent spread through travel. Recently, the emergence of *C. auris* in Iran identified a fifth genetically distinct clade, albeit with only a single isolate identified to date [21]. The important virulence factors include the ability to form biofilms [22,23], the production of phospholipases and proteinases [24], the propensity to colonize patients and their environments for weeks to months, resulting in efficient transmission in healthcare settings [25,26], and resistance to antifungals. Most isolates studied had high-level fluconazole resistance (minimum inhibitory concentrations, MICs > 64 µg/mL). Additionally, up to 30% of isolates demonstrated reduced susceptibility to amphotericin B. Furthermore, 5% can be resistant to the echinocandins, which are the currently recommended first-line antifungals for candidemia [20,27]. Rudramurthy et al. performed a retrospective study to identify risk factors for acquisition *C. auris* candidemia in intensive care unit patients compared to other *Candida* species. Most importantly, prior antifungal use, vascular surgery, admission to a public-sector hospital, underlying pulmonary disease and indwelling urinary catheterization were associated with *C. auris* infection [28]. Despite being implicated in a minority of candidemia cases, the mortality associated with *C. auris* has been reported as up to 60% in some studies [29]. Collectively, these epidemiologic shifts restrict the utility of fluconazole as empiric therapy in candidemia.

### 3.3. Cryptic Speciation in Cryptococcus

For decades, after its description at the end of the 19th century, *C. neoformans* was a rare cause of human disease. However, the recognition of meningoencephalitis and other infections caused by this opportunistic organism increased in light of the AIDS epidemic in the 1980s and 1990s. Another species, *C. gattii*, has become an important cause of disease in both immunocompromised and immunocompotent hosts with geographic tropism, notably the Pacific Northwest coast of North America [39]. 

Based on molecular typing, Hagen et al. proposed that the genus be further subdivided into a number of new species [40]. *Cryptococcus neoformans* sensu stricto (formerly *C. neoformans* serotype A) differs in primarily causing meningoencephalitis rather than skin lesions and in geographical distribution [41]. *Cryptococcus deneoformans*, previously known as *C. neoformans* serotype D or *C. neoformans* var. *neoformans*, has been shown to occur at higher rates in HIV-infected individuals over the age of 60, those with cutaneous manifestations, and those who use intravenous drugs. The rates of infection were lower in Africans [41]. *Cryptococcus gattii* sensu stricto primarily affects apparently healthy individuals, with rates highest in Australia. *Cryptococcus bacillisporus* more commonly affects HIV-infected and immunocompromised hosts and the global numbers are too small to propose a geographic distribution. *Cryptococcus deuterogattii* is associated with the previous outbreak in Vancouver Island and Pacific Northwest of the United States. *C. deuterogattii* has higher MICs for isavuconazole compared to other species. *Cryptococcus tetragattii* comprises the majority of cases of infections with *C. gattii* species complex in Africa. However, little is known of the epidemiology of another species, *C. decagattii*.

## 4. Emerging Mould Infections

### 4.1. Emergence of Azole-Resistance in Aspergillus fumigatus

Invasive aspergillosis (IA) carries a high fatality rate in immunocompromised patients. *Aspergillus fumigatus,* the most common causative species, has been almost universally susceptible to newer generation triazole antifungals, such as itraconazole, voriconazole and posaconazole. Under the pressure of pervasive antifungal use, however, resistance has become an emerging clinical challenge. The primary target of medical triazoles is lanosterol 14α-demethylase, which is essential for ergosterol biosynthesis and fungal cell membrane stability. The mutations in the *cyp51A* and *cyp51B* genes, which encode this demethylase, are important mechanisms of azole resistance in *Aspergillus* species. Among the most common *cyp51A* mutations conferring azole resistance is TR_34_/L98H—a 34-bp tandem repeat in the gene’s promoter region with an associated substitution of lysine to histidine at codon 98, which results in an eight-fold upregulation of lanosterol 14α-demethylase [42,43]. Another leading *cyp51A* mutation associated with azole resistance in *A. fumigatus* is TR_46_/Y121F/T289A, which is a 46-bp tandem repeat with substitution of tyrosine to phenylalanine and threonine to alanine at codons 121 and 289, respectively. Recently, Rybak et al. showed that while *cyp51A* mutations were insufficient on their own to confer azole resistance, a newly described mutation at HMG reductase gene *hmg1* can cause pan-azole resistance in clinical isolates [44]. 

In the late 1990s, several patients were found with itraconazole-resistant *A. fumigatus* in the United States, which was suspected to have developed as the result of prolonged antimicrobial therapy [45]. In the following decade, multiple European centers described increasing azole resistance in clinical *A. fumigatus* isolates, including from patients who had not previously been treated with azoles and environmental isolates. In the Netherlands, a nationwide surveillance of *A. fumigatus* specimens isolated between 1945 and 1998 reported that fewer than 2% of isolates were highly resistant to itraconazole and no isolates were voriconazole resistant [46]. In 2016, the rate of azole resistance increased to 5.3% [47,48]. This increase occurred in parallel with the widespread use of agricultural demethylase inhibitors throughout Europe [49,50,51,52]. Moreover, the repeated exposure of susceptible *A. fumigatus* strains to azoles used as pesticides can induce genetic mechanisms identical to those observed in patients and that select for resistance to medical azoles [53]. Azole-resistant *A. fumigatus* is also becoming recognized beyond Europe, and is now reported from six continents [54]. In the United States, the most recent estimate of azole resistance in *A. fumigatus* isolates was 1.5%. Furthermore, 30% of these isolates did not have an identifiable *cyp51A* gene mutation, necessitating further characterization of other loci of resistance [55]. The global distribution of countries reporting azole-resistant *A. fumigatus* is shown in Figure 2.

However, less is known about antifungal resistance mechanisms and patterns in non-fumigatus *Aspergillus* species. Notably, *A. flavus*, the second most common cause of aspergillosis [98,99,100], has also shown low rates of azole resistance in small surveillance studies. Some studies from India, the Netherlands, and China quote azole resistance rates of up to 5% [101,102,103,104]. The consequences of *A. flavus* azole resistance may be dire in tropical and semi-arid climates, such as in India, South America and Saudi Arabia, where absolute rates of *A. flavus* infections are higher compared to North America and Europe. The mechanisms of azole resistance in *A. flavus* go beyond that of *cyp51* gene mutations, with some studies showing a role of overexpression of several loci generating multidrug efflux pumps [102,105].

### 4.2. Influenza-Associated Invasive Pulmonary Aspergillosis

Recently, several studies from the Netherlands and Belgium have reported IA to be a common complication of severe influenza [106,107]. From 2015 to 2016, 23 cases of influenza-associated IA were diagnosed in Dutch intensive care units (ICUs) and half of these patients had no other risk factor for invasive fungal diseases [108]. A subsequent retrospective study reviewed 630 admissions of patients to seven ICUs from 2009 to 2016 with severe community acquired pneumonia. Half of the patients had influenza pneumonia, the presence of which increased the risk of developing IA from 5% to 14% [109]. Immunocompetent patients with influenza-associated aspergillosis had almost 50% mortality in the three months following diagnosis, compared to 29% in matched ICU patients with severe influenza but without IA. This study demonstrated influenza infection to be an independent risk factor for IA. Schauwvlieghe et al. hypothesized that the underlying pathophysiology relies on the influenza virus severely damaging the respiratory epithelium and mucociliary clearance function to allow *Aspergillus* invasion [109]. Further studies are underway to assess the role of antifungal prophylaxis in patients with severe influenza pneumonia.

### 4.3. Non-Aspergillus Mould Infections

The use of antifungal prophylaxis in immunosuppressed individuals has also led to an increase of invasive mould infections with non-*Aspergillus* species. In 2012, Auberger et al. studied the breakthrough invasive fungal infections in patients with prolonged neutropenia receiving posaconazole [110]. They noted an increase in breakthrough Mucorales infections. More recent studies from India suggest *Apophysomyces* species are becoming more prevalent causes of mucormycosis and are associated with hospital-acquired cutaneous infections [111,112].

More recently, increases of infections with *Lomentospora prolificans* (formerly *Scedosporium prolificans*) and *Fusarium* species, which are more often resistant to voriconazole and posaconazole, have also been observed in immunosuppressed hosts [113]. In a single-center study, Dalyan Cilo et al. showed that over the 20-year period of 1995–2014, cases of fusariosis increased from an average of 0.67 cases per year to 4.8 cases per year. This increase was associated with both a rise in locally invasive and disseminated infections. The predominant species implicated was *F. proliferatum* [114]. A multicenter study from seven European nations identified 76 cases of fusariosis over a five-year period. The MICs of isolates were generally low for amphotericin B, but variable for azoles (with the highest MICs noted for itraconazole) [115]. Seidel et al. reported on 208 cases of scedosporiosis and 56 cases of lomentosporiosis in the literature and Fungiscope®, a 74-nation registry of rare invasive fungal diseases. Almost half of the cases of *Scedosporium* spp. infections occurred in immunocompetent hosts, whereas 70% of *Lomentospora prolificans* infections occurred in patients with immunocompromising conditions, mostly malignancy or solid organ transplantation (SOT) [116]. *Lomentospora prolificans* isolates were pan-resistant to virtually all systemically active antifungals, including azoles, terbinafine and amphotericin B [116]. 

In SOT and haematopoietic stem cell transplant (HSCT) recipients, non-*Aspergillus* moulds are an emerging cause of late-onset invasive fungal infection. Between 2001 to 2006, the Transplant Associated Infection Surveillance Network (TRANSNET) identified 56 cases of phaeohyphomycosis [117] and 105 cases of mucormycosis [98,99], and 31 cases of fusariosis [98] in transplantation recipients in the United States. Non-*Aspergillus* mould infections tended to occur later following transplantation than IA [98,99]. From 2004 to 2007, Neofytos et al. highlighted that up to 15% of invasive mould infections following transplant were non-*Aspergillus* [118]. Among HSCT recipients, 29 cases of invasive *L. prolificans* infection have been published and associated with over 80% mortality [119].

### 4.4. Indian Epidemic of Resistant Dermatophytosis

A shift in the epidemiology and microbiology of dermatophytoses in India has occurred over the last 20 years. Previously, *Trichophyton rubrum* has been the prevalent pathogen. However, there has now been an emergence of *T. mentagrophytes* infection throughout the country [120,121]. This change is substantial, with *T. mentagrophytes* rising from 20% of dermatophytoses to over 90% in less than 15 years [121]. The sequencing of the internal transcribed spacer region of the ribosomal DNA has determined that this strain is genetically distinct from other reference strains worldwide [121]. This epidemic resulted in more efficient human-to-human transmission and lesions that were more inflammatory and eruptive than previous, with a predilection for the face [121,122]. In one study, 78% of skin lesions presenting to an Indian hospital were a dermatophytosis, which was well above the 20–25% worldwide prevalence [123]. 

The development of resistance among these fungi to terbinafine is troubling. Terbinafine is the oral and topical squalene epoxidase inhibitor that has long been the first line of therapy for superficial dermatophytosis. In Japan, 1% of *Trichophyton* isolates harbored point mutations in the squalene epoxidase gene (*SQLE*) conferring terbinafine resistance [124,125]. The increasing rates of dermatophytosis and emerging antifungal resistance is thought to be due to the unregulated use of antifungals, as well as fixed drug combination creams containing steroids, antifungals, and antibacterials, which have been used for many undifferentiated skin lesions in India [120].

## 5. Emerging Dimorphic Fungal Infections

### 5.1. Emergence of Distinct, Novel Pathogens in Emergomyces and Blastomyces, and Cryptic Speciation in Histoplasma

Over the past five years, a taxonomic overhaul of medically important, dimorphic fungi in the family Ajellomycetaceae has occurred [126]. In 2013, Kenyon et al. reported disseminated disease in South African patients with advanced HIV caused by a previously unrecognized dimorphic pathogen, which the authors classified as a likely *Emmonsia* species on the basis of relatedness to *Emmonsia pasteuriana*, which was then an obscure fungus reported only once previously [127]. Schwartz et al. later reported an additional 39 cases of disseminated disease in immunocompromised patients from around South Africa [128]. Nearly all the patients had skin lesions, and pulmonary disease was also common. The patients were frequently misdiagnosed as tuberculosis, and in a quarter of cases, a diagnosis of fungal disease only was made when ante-mortum blood cultures grew a mould after the patients had died. In total, half of all patients died. After these reports, additional *Emmonsia*-like isolates were discovered in several global collections [129]. In 2017, several *Emmonsia*-like fungi were re-classified into the newly formed genus *Emergomyces*. Currently, five *Emergomyces* species have been recognized with cases reported from four continents [2] (Figure 3). *Emergomyces pasteurianus* infections have been reported from Italy, Spain, France, the Netherlands, India, China, South Africa, and Uganda [130]. In North America, *Es. canadensis* has been implicated in four cases of disseminated mycoses in immunocompromised patients [131]. One case of disseminated *Es. orientalis* infection has been described in a previously healthy man in China [132]. *Emergomyces africanus* has been implicated in dozens of cases of disseminated disease in patients with advanced HIV disease [128] and portends a significant mortality [133]. Further, in vitro data suggest resistance to fluconazole is common, but susceptibility to other triazoles and amphotericin B is generally preserved [134,135]. 

Although *Blastomyces* was first described over a century ago, a careful observation of mycologists supplemented by advances in molecular characterization of fungi has led to new appreciations of the genetic diversity and in the geographic niches of these fungi [2]. Until recently, the only known agent of blastomycosis in North America was *B. dermatitidis*, which is endemic to Mid-Eastern Canada and the United States [140]. Over the last decade, taxonomic changes have led to a broadening of the genus, which now includes at least three additional, clinically relevant species that differ in genetics, virulence, and geographic distribution.

From building on the work of others [141], Meece et al. from Wisconsin demonstrated the genetic variation within clinical and environmental isolates of *B. dermatitidis* from North America. In 2011, these authors recognized the existence of two distinct genetic populations within *B. dermatitidis* [142]. Soon thereafter, Brown et al. described one of these subpopulations as a cryptic species, *B. gilchristii* [143]. While phenotypic and clinical distinctions may exist, they have not yet been convincingly shown. There do seem to be geographic differences, though many if not most regions of geographic risk for blastomycosis are sympatric for both *B. gilchristii* and *B. dermatitidis* (herein referred to collectively as *B. dermatitidis* species complex) [143,144,145]. 

Retrospectively, at least some cases of blastomycosis occurring in Canada and the United States can be attributed to *B. helicus* (formerly *Emmonsia helica*). The earliest known case occurred in 1970 but was only recently reclassified. These infections differ from those caused by *B. dermatitidis* species complex in their more western North American distribution (Figure 4) through their predilection for immunocompromised, rather than immunocompetent hosts, increased incidence of fungemia and systemic dissemination, and possibly higher rates of infection in felines than observed with *B. dermatitidis* species complex [146].

Blastomycosis has been reported outside of North America, in patients not known to have traveled to areas of geographic risk. Most cases of blastomycosis reportedly acquired outside of North America have been in Africa [147], where locally-acquired infections have been reported from over two-dozen African countries (Schwartz IS, unpublished data). Other cases of blastomycosis have been reported from the Middle East, and as far east as India. At least some of these cases outside of North America are due to one or more distinct species [2,126,147]. Dukik et al. first described *B. percursus* in 2015 from isolates obtained from patients with extrapulmonary blastomycosis from South Africa and Israel. This species differed from *B. dermatitidis* species complex not only in geographic distribution, but also in the pattern of conidia production [126]. 

The phylogenetics of *Histoplasma* was revisited by Sepúlveda et al. who noted genetically distinct clades with geographic variation. These authors proposed that *H. capsulatum* sensu stricto should be the name for the taxa located in Panama where histoplasmosis was first described, while *H. mississippiense*, *H. ohiense*, and *H. suramericanum* were proposed for the other three American taxa found mostly around the Mississippi River, Ohio River, and South America, respectively [148]. The clinical relevance of these distinctions has not been established in humans. In mouse models, *H. suramericanum* produced acute granulomatous necrotizing lung inflammation associated with high mortality, while *H. ohiense* and *H. mississippiense* were associated with chronic lung disease [149]. This taxonomic divergence remains controversial and is not yet widely accepted. On the other hand, *H. duboisii*, which causes histoplasmosis in western Africa, has long been appreciated to differ by the production of larger yeast cells and for causing predominantly cutaneous and osteo-articular manifestations [148,150,151]. 

### 5.2. Shifting Areas of Geographic Risks: Blastomycosis, Coccidioidomycosis, and Histoplasmosis

The three most common geographically restricted, dimorphic fungal infections (endemic mycoses) in North America are coccidioidomycosis, histoplasmosis and blastomycosis. The causative fungi—*Coccidioides* spp., *Histoplasma* spp., and *Blastomyces* spp., respectively—exist as moulds in the environment. The aerosolized spores can become inhaled by mammals, which lead to a temperature-dependant transformation in mammalian tissue to yeast-like cells. The suitable environmental conditions to support the mould phases are required for the risk of autochthonous infection, although the precise environmental factors remain poorly understood [152]. 

Recent data suggested that the geographic ranges of these fungi have expanded. The region of geographic risk for blastomycosis in North America is generally considered to include states and provinces adjacent to the Great Lakes and the St. Lawrence, Ohio and Mississippi Rivers, but recent case series have suggested that *B. dermatitidis* species complex can be found as far west as Saskatchewan [153] and as far east as New York [154]. Histoplasmosis, classically most associated in North America with the Ohio River Valley, has been reportedly acquired in Montana [155] and even Alberta [156], areas not previously considered endemic. Coccidioidomycosis, in the United States classically found in south-western states, has been acquired in Washington state, where *C. immitis* has been isolated from soil [157]. The precise reasons for the apparent expansions of the geographic ranges for these diseases are unknown but may relate to climate change [157,158]. 

### 5.3. The Proliferation of Cases of Zoonotic Sporotrichosis in Brazil

Originally described in 1898, sporotrichosis is a chronic granulomatous infection with a worldwide distribution caused by the thermally dimorphic fungus *Sporothrix* spp. Until the early 2000s, *S. schenckii* was the only known species implicated in the disease, but genetic analyses have led to the description of additional species, including *S. globosa*, *S. mexicana*, *S. luriei* and *S. brasiliensis* [159,160]. *Sporothrix brasiliensis* is a particularly virulent species capable of producing profound inflammatory responses [161,162]. Furthermore, unlike other causes of sporotrichosis, *S. brasiliensis* is associated with zoonotic (rather than sapronotic) transmission involving cats (Figure 5) [163]. These factors have led to outbreaks of sporotrichosis amongst feline and human populations in Brazil over the last two decades [164]. Over the 12-year period of 1987 to 1998, only 13 cases of human sporotrichosis were described in Brazil [165]. These rates have grown, especially in the state of Rio Grande do Sul, which has seen a four-fold increase of *S. brasiliensis* infections in humans from 2010 to 2014 [166] and the city Rio de Janeiro, which is hyperendemic, possibly due to overcrowding [167]. Between 2012 and 2017, 101 human cases of zoonotic sporotrichosis were reported [168].

## Figures and Tables

**Figure 1 jof-05-00067-f001:**
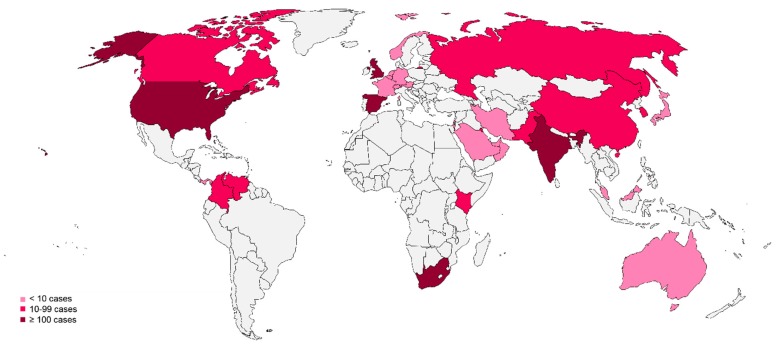
Countries reporting cases of *Candida auris* through June 15, 2019 [19,30,31,32,33,34,35,36,37,38].

**Figure 2 jof-05-00067-f002:**
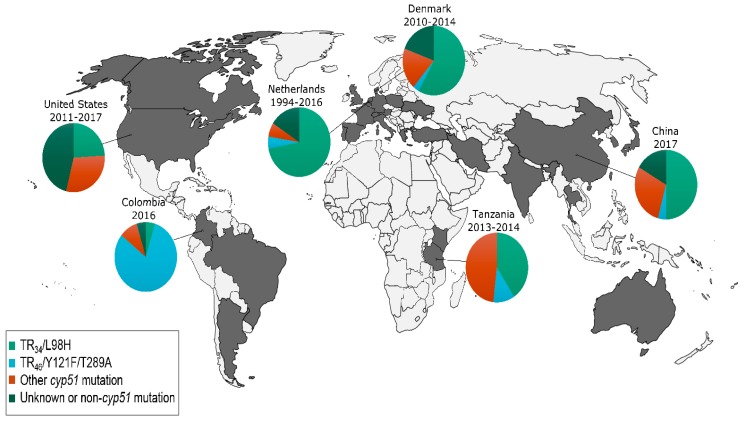
Countries reporting the presence of azole-resistant *Aspergillus fumigatus* and mechanisms of resistance in surveyed isolates [47,48,49,52,55,56,57,58,59,60,61,62,63,64,65,66,67,68,69,70,71,72,73,74,75,76,77,78,79,80,81,82,83,84,85,86,87,88,89,90,91,92,93,94,95,96,97].

**Figure 3 jof-05-00067-f003:**
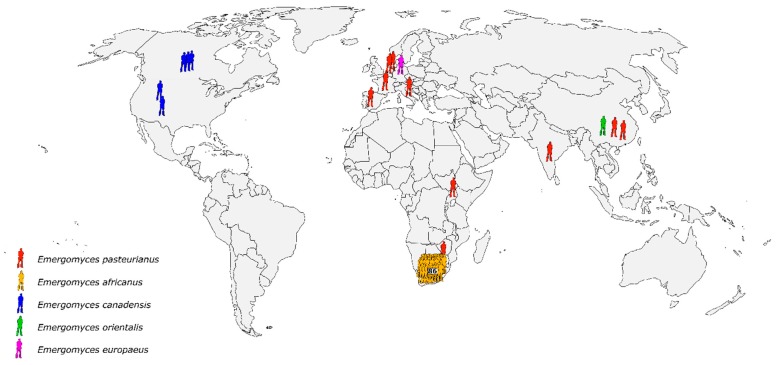
Global distribution of reported cases of emergomycosis [2,130,131,132,136,137,138,139].

**Figure 4 jof-05-00067-f004:**
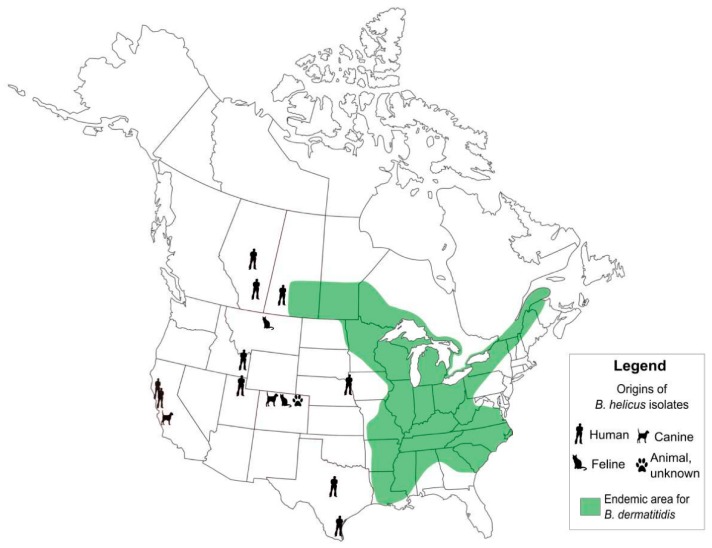
The distribution of human and veterinary cases of disease caused by *Blastomyces helicus* in relation to the classic region of geographic risk for *B. dermatitidis* species complex. Reproduced with permission from [146].

**Figure 5 jof-05-00067-f005:**
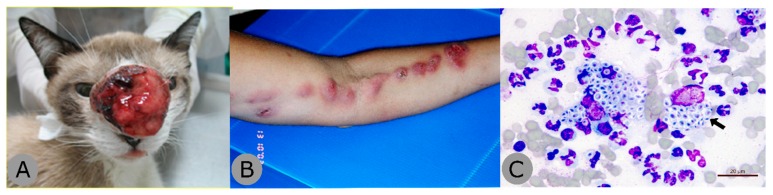
Zoonotic sporotrichosis caused by *Sporothrix brasiliensis*. (**A**) Ulcerative tumour-like lesion on an infected cat; (**B**) lymphocutaneous zoonotic sporotrichosis in a human patient acquired from a cat; (**C**) yeast-like cells of *S. brasiliensis* (arrow) seen in an impression smear of a feline skin ulcer on a glass slide, Quick Panoptic stain (Instant Prov Kit; Newprov, Pinhais, Brazil). Images courtesy of Rodrigo Menezes, LAPCLIN-DERMZOO, Evandro Chagas National Institute of Infectious Diseases (INI), Oswaldo Cruz Foundation (Fiocruz), Rio de Janeiro, Brazil.

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
