# Peer review of "Emerging Fungal Infections: New Patients, New Patterns, and New Pathogens"

_jof, 2019, doi:10.3390/jof5030067_

Round 1
Reviewer 1 Report
The review article by Daniel Z.P. Friedman and Ilan S. Schwartz is one of the outstanding articles, covering the most recent epidemiological occurrence of many fungal infections, especially emerging, worldwide. With the advent of medical interventions, further 'wounded' by global travel, has led to several fungal, were often innocuous, and the emerging & new species, infections becoming deadly and anti-fungal resistant. It is imperative to understand the global occurrence and emergence of fungal infections, which require long-term therapy complicated by emergence of resistance. The review article gives clinicians as well as researchers to come up with suitable and new modalities to overcome these emerging diseases including developing effective vaccines and novel therapeutics.
Author Response
Re: jof-541133
Dear Drs. Chiller and Jackson,
We wish to thank the editors and anonymous reviewers for the constructive feedback on our manuscript, “Emerging Fungal Infections: New Patients, New Patterns, and New Pathogens”.
We are grateful for the opportunity to improve and resubmit this work.
There were no comments necessitating revision from Reviewer 1. Below, we address Reviewer 2’s comments point-by-point:
1. I suggest to the authors to check the very recent paper by Stavrou and colleagues (FEMS Yeast Research) regarding shifts in Candida prevalence, which might also be due to different diagnostic approaches.
This has been addressed by the addition of lines 93-100 reflect findings from this study.
2. The moulds-section starts with Aspergillus fumigatus and the observed resistance mechanisms. Then the text smoothly goes to non-Aspergillus mould infections, but authors forgot to spend some sentences about Aspergillus flavus infections and the emergence of resistance.
Although the focus was intended to be on A. fumigatus, we have added a few lines to address resistance with A. flavus (lines 299-334). Another review in this issue addresses A. flavus resistance in more detail.
3. In line 238-242 the genus is abbreviated to "Es." while it can remain "E."
We retain use of Es. for Emergomyces to dinstinguish from Emmonsia (Ea.).
4. The Emergomyces-part is, contrary to other sections, finished with a part about treatment of Emergomyces-infections. Any reference for that?
To maintain consistency, treatment of Emergomyces was removed.
5. In line 263-271 the description of B. gilchristii is mentioned, but that this seems to be not convincing. How can the two species reliably identified/discriminated from each other?
To date, studies have only shown genetic and geographic differences between B. dermatidis and B. gilchristii. No clinical or phenotypic differences have been described conclusively.
6. In the Blastomyces-section there are some references expected, but currently missing (sentences "where....(unpublished data)" and "At least...distinct species."
References were added when available. The sentences now reads “…where locally-acquired infections have been reported from over two-dozen African countries (Schwartz IS, unpublished data)” and “At least some of these cases outside of North America are due to one or more distinct species[2,126,147].”
7. The part about Coccidioides and Histoplasma is short, too short in my humble opinion, especially given the fact that Histoplasma has recently (Sepulveda et al., 2017) been split into several species, partly based on different clinical presentations and outcome. Coccidioides immitis is mentioned, but C. posadasii not, why? The Coccidioides/Histoplasma-section has a very strong focus on North America.
The intent of this section was to address the changing distribution of known species. We have, however, added some comments on the new species in the genus Histoplasma (line 620-648)
8. The major fungal genera are highlighted in this review, except for Cryptococcus while it fits into the scope of this review, its taxonomy has been revised several years ago, there are numerous reports of shift in patient-populations (e.g. C. neoformans s.s. from HIV to commonly found as culprit of disease among elderly/apparently immunocompetent), differences in in vitro antifungal susceptibility among cryptococcal species, different patient populations (e.g. C. deneoformans significantly related to cutaneous infections; infections in the elderly). What was the reason to exclude Cryptococcus?
This has been addressed with the addition of lines 150-170.

Reviewer 2 Report
With this review-manuscript authors summarize the current state of fungal epidemiology that has changed during the past decade(s) due to revised taxonomy and new patient populations, but more importantly the emergence of resistance. After a brief introduction of what has been changed in terms of patient population, authors move to the different pathogens. The first section deals with Candida and the shift in species-prevalence, which is mainly on C. glabrata and parapsilosis while a large paragraph is providing information about Candida auris. I suggest to the authors to check the very recent paper by Stavrou and colleagues (FEMS Yeast Research) regarding shifts in Candida prevalence, which might also be due to different diagnostic approaches. Authors refer to Lockhart et al regarding the identification of different geographic populations of C.auris, but that WGS study confirms what previous studies found by applying different molecular techniques. The moulds-section starts with Aspergillus fumigatus and the observed resistance mechanisms. Then the text smoothly goes to non-Aspergillus mould infections, but authors forgot to spend some sentences about Aspergillus flavus infections and the emergence of resistance. The recent reports of terbinafine resistant dermatophytes is highlighted, there is an unexplained comment in this part namely "a finding that the authors found surprisingly high given the size of the fungus' genome". Question is why is to surprising? Small genome or large genomes, microbes develop resistance... In line 238-242 the genus is abbreviated to "Es." while it can remain "E.". The Emergomyces-part is, contrary to other sections, finished with a part about treatment of Emergomyces-infections. Any reference for that? In line 263-271 the description of B. gilchristii is mentioned, but that this seems to be not convincing. How can the two species reliably identified/discriminated from each other? In the Blastomyces-section there are some references expected, but currently missing (sentences "where....(unpublished data)" and "At least...distinct species.". The part about Coccidioides and Histoplasma is short, too short in my humble opinion, especially given the fact that Histoplasma has recently (Sepulveda et al., 2017) been split into several species, partly based on different clinical presentations and outcome. Coccidioides immitis is mentioned, but C. posadasii not, why? The Coccidioides/Histoplasma-section has a very strong focus on North America. The major fungal genera are highlighted in this review, except for Cryptococcus while it fits into the scope of this review, its taxonomy has been revised several years ago, there are numerous reports of shift in patient-populations (e.g. C. neoformans s.s. from HIV to commonly found as culprit of disease among elderly/apparently immunocompetent), differences in in vitro antifungal susceptibility among cryptococcal species, different patient populations (e.g. C. deneoformans significantly related to cutaneous infections; infections in the elderly). What was the reason to exclude Cryptococcus?
Author Response

(The authors gave the same response as above.)
